# Preclinical Repurposing of Sitagliptin as a Drug Candidate for Colorectal Cancer by Targeting *CD24*/*CTNNB1*/*SOX4*-Centered Signaling Hub

**DOI:** 10.3390/ijms25010609

**Published:** 2024-01-03

**Authors:** Jing-Wen Shih, Alexander T. H. Wu, Ntlotlang Mokgautsi, Po-Li Wei, Yan-Jiun Huang

**Affiliations:** 1Ph.D. Program for Cancer Molecular Biology and Drug Discovery, College of Medical Science and Technology, Taipei Medical University and Academia Sinica, Taipei 11031, Taiwan; shihjw@tmu.edu.tw (J.-W.S.); d621108006@tmu.edu.tw (N.M.); 2Graduate Institute of Cancer Biology and Drug Discovery, College of Medical Science and Technology, Taipei Medical University, Taipei 11031, Taiwan; 3TMU Research Center of Cancer Translational Medicine, Taipei Medical University, Taipei 11031, Taiwan; 4The Ph.D. Program for Translational Medicine, College of Medical Science and Technology, Taipei Medical University, Taipei 11031, Taiwan; chaw1211@tmu.edu.tw; 5International Ph.D. Program for Translational Science, College of Medical Science and Technology, Taipei Medical University, Taipei 11031, Taiwan; 6Clinical Research Center, Taipei Medical University Hospital, Taipei Medical University, Taipei 11031, Taiwan; 7Graduate Institute of Medical Sciences, National Defense Medical Center, Taipei 114, Taiwan; 8Division of Colorectal Surgery, Department of Surgery, Taipei Medical University Hospital, Taipei Medical University, Taipei 110, Taiwan; poliwei@tmu.edu.tw; 9Department of Surgery, School of Medicine, College of Medicine, Taipei Medical University, Taipei 110, Taiwan; 10Division of General Surgery, Department of Surgery, Taipei Medical University Hospital, Taipei Medical University, Taipei 110, Taiwan

**Keywords:** colorectal cancer (CRC), cancer stem cells (CSCs), cancer-associated fibroblasts (CAFs), sitagliptin, CSC inhibitor

## Abstract

Despite significant advances in treatment modalities, colorectal cancer (CRC) remains a poorly understood and highly lethal malignancy worldwide. Cancer stem cells (CSCs) and the tumor microenvironment (TME) have been shown to play critical roles in initiating and promoting CRC progression, metastasis, and treatment resistance. Therefore, a better understanding of the underlying mechanisms contributing to the generation and maintenance of CSCs is crucial to developing CSC-specific therapeutics and improving the current standard of care for CRC patients. To this end, we used a bioinformatics approach to identify increased *CD24*/*SOX4* expression in CRC samples associated with poor prognosis. We also discovered a novel population of tumor-infiltrating *CD24*+ cancer-associated fibroblasts (CAFs), suggesting that the *CD24*/*SOX4*-centered signaling hub could be a potential therapeutic target. Pathway networking analysis revealed a connection between the *CD24*/*SOX4*-centered signaling, *β-catenin*, and *DPP4*. Emerging evidence indicates that *DPP4* plays a role in CRC initiation and progression, implicating its involvement in generating CSCs. Based on these bioinformatics data, we investigated whether sitagliptin, a *DPP4* inhibitor and diabetic drug, could be repurposed to inhibit colon CSCs. Using a molecular docking approach, we demonstrated that sitagliptin targeted *CD24*/*SOX4*-centered signaling molecules with high affinity. In vitro experimental data showed that sitagliptin treatment suppressed CRC tumorigenic properties and worked in synergy with 5FU and this study thus provided preclinical evidence to support the alternative use of sitagliptin for treating CRC.

## 1. Introduction

Colorectal cancer (CRC) is the third most prevalent cancer, leading to high mortality in women and men worldwide [1,2]. Despite the advances in surgical procedures, systemic treatments, overall survival, and disease-free survival in patients with advanced metastatic CRC remain poor [3,4]. Distant metastasis and the development of treatment resistance have been attributed to a subpopulation of CRC cells, commonly referred to as cancer stem cells (CSCs) [5,6]. CSCs are defined as possessing unlimited cell division and self-renewal ability and are characterized by expressing *CD24CD24*, *CD44*, *CD133*, *OCT4*, aldehyde dehydrogenase 1 (*ALDH1*), and CXC-chemokine receptor 4 (*CXCR4*) [7]. The presence and emergence of CSCs are associated with disease progression, treatment failure, and relapse. Studies have indicated that targeting and eliminating CSCs is a rational and potential venue for achieving the complete remission of CRC [8]. However, the CRC tumor core and tumor microenvironment (TME) are highly heterogeneous and complex, presenting a daunting task for CSC-specific therapeutic development. Thus, a better understanding of CSC biology and its interactions with the TME is essential.

Vigorous efforts have been invested in identifying CSC markers, and recently, dipeptidyl-peptidase 4 (*DPP4*, also known as *CD26*) has been reported as a marker of stemness (CSCs) in several cancers and has been associated with distant metastasis [9,10,11,12]. *DPP4* is a multifunctional cell surface protein reported to play a pivotal role in various cancer types, including CRC, through its interaction with the tumor microenvironment and is widely expressed in the T-lymphocytes [13,14,15]. When upregulated in CRC, *DPP4* was shown to promote cancer development, progression, and metastasis, making it a potential and novel therapeutic target for CRC [16,17,18]. Notably, *DPP4* inhibitors (DPPi) have been developed and safely used for treating type II diabetes. This discovery prompted researchers to investigate *DPP4*i’s potential usage for treating cancer patients.

Multiple studies have shown that approximately 70% of CRC patients exhibit high expression levels of *CD24*, which has also been demonstrated to possess CSC properties [19,20]. Others have shown that the genetic alterations on *β-catenin* prompt the *β-catenin*/T cell transcription factor (TCF) signaling in CRC [21,22]. *β-catenin*/*TCF4* pathway at the transcriptional level is regulated by *SOX4* [23]. *SOX4* has recently been identified in various cancer types and implicated in cancer initiation. However, little is known about its involvement in CSCs. These observations suggest molecular connections between *CD24*/*SOX4*-centered signaling and *DPP4*/*β-catenin*, making them potential therapeutic targets. Gene association and enrichment analyses showed that *CD24*/*SOX4*-centered signaling hub was associated with cancer cell metabolism, immune responses, ECM regulation, and *Wnt*/*β-catenin* pathways. Based on these preliminary data, we aim to provide preclinical support for sitagliptin (a *DPP4* inhibitor) as an inhibitor of CSC in CRC.

## 2. Results

### 2.1. Identification of Differentially Expressed Genes (DEGs) in CRC

Differentially expressed genes (DEGs) from four (4) GEO microarray datasets of CRC patients (accession numbers GSE21510, GSE32323, GSE110223, and GSE110224) were selected based on the obtained *p*-values (<0.05), and cut-off values. The expression data based on the number of patients and number of expressed genes, including upregulated and downregulated genes, are represented in volcano plots (Figure 1A–D); moreover, the 59 overlapping upregulated DEGs can be obtained from the constructed Venn diagram (Figure 1E). Furthermore, all the overexpressed genes obtained from overlapping genes are presented on a heatmap diagram (Figure 1F).

### 2.2. CD24/CTNNB1/SOX4 Are Overexpressed in Colorectal Cancer (CRC)

The 59 overlapping genes were obtained from microarray datasets, including *CD24*, *CTNNB1*, and *SOX4* oncogenes. To validate this, we explored GEPIA2 online bioinformatics tool with default settings. As anticipated, the obtained results showed that the mRNA levels of *CD24*/*CTNNB1*/*SOX4* oncogenes were upregulated in CRC tissues as compared with normal tissues (Figure 2A–C). Moreover, we used the HPA as an independent tool to validate the expression of *CD24*/*CTNNB1*/*SOX4* in CRC tissue, and the results revealed the high expression levels of these oncogenes in CRC tissues as compared with normal tissues, with medium staining intensities displayed on each tissue, and *p* < 0.05 being considered statistically significant (Figure 2D–I). For further analysis, we explored the cBioportal tool to predict the co-expression of *CD24*/*CTNNB1*/*SOX4* in CRC. Based on the analysis, *CD24* with *SOX4*, *CTNNB1* with *CD24*, and *SOX4* with *CD24* were found to be co-expressing in CRC, with positive Spearman and Pearson values and a *p*-value < 0.05 considered significant (Figure 2J–L), with *p* < 0.05 considered statistically significant.

### 2.3. Increased CD24/CTNNB1/SOX4 Expression Is Associated with Poor Prognosis in Colon Cancer Patients

We first searched and analyzed a dataset containing 12 CRC patients’ samples (66,050 cells using a 10× genomics platform) [24] (Figure 3A). We first explored an established cancer stemness marker, *CD24* (Figure 3B) and an EMT marker for *SOX4* expression (Figure 3C). *CD24* expression was predominantly in the malignant and epithelial cell clusters. *CD24*/*CTNNB1*/*SOX4* expression was detected in multiple cell clusters, including malignant, endothelial, epithelial, fibroblast, myofibroblast, and DC cell clusters (Figure 3D). We then queried the TCGA datasets using GEPIA2 software. We found that increased *CD24*/*CTNNB1*/*SOX4* expression was associated with shorter overall survival (Figure 3E,F).

### 2.4. Connecting DPP4 with CD24/CTNNB1/SOX4-Centered Signaling Network

Next, we explored the signaling pathways associated with *CD24*/*SOX4* using the STRING database under high confidence (with a minimal interaction score of 0.700). Our analyzed results revealed the correlation among *DPP4*/*CD24*/*CTNNB1*/*SOX4* expression. The protein interaction analysis displayed the following network statistics: number of nodes (4), number of edges (5), average node degree (2.5), average local clustering coefficient (0.8), and PPI enrichment *p*-value (8.97 × 10^−5^), and the confidence cut-off value representing interaction links was adjusted to the highest, 0.900 (Figure 4A). Next, we utilized the DAVID and FunRich programs to determine the gene ontology (GO), which involved (6) biological functions (BP), including immune response (12.5%), cell communication (37.5%), protein metabolism (25%), signaling transduction (37.5%), chromosome segregation (12.5%), and the regulation of cell cycles (12.5%) (Figure 4B). Furthermore, the (6) KEGG pathways were also shown: regulation of nuclear *β-catenin* (71.4%), E-cadherin signaling event (85.7%), stabilization and expansion of E-cadherin (85.7%), canonical *Wnt* signaling pathway (71.4%), noncanonical *Wnt* signaling pathway (71.4%), and E-cadherin signaling in the nascent adherens junction (85.7%) (Figure 4C). Collectively, the gene enrichment analyses strongly suggested that *CD24*/*SOX4* connected with *Wnt*/*β-catenin* signaling networks and participated in cellular communication, immune/metabolism regulation, and ECM remodeling.

### 2.5. Heterogeneity of CRC Cancer-Associated Fibroblasts (CAFs) and Exploration of CD24+ CAFs in CRC

A seminal study by Qi et al. indicated that heterogenous CAF populations infiltrate the CRC tumor microenvironment [25]. One subtype of CRC-infiltrating CAFs was *CD24*+ (the first column denoted by the red asterisk, Figure 5A). High expression levels of *COL1A1* and *COL3A* characterize these *CD24*+ CAFs. Further analyses classified different CAF-associated gene sets by transcriptional factor (TF) regulon. *CD24*+ CAFs were found which were most significantly associated with TF regulon: *POU5F1 (OCT6)*, *PKNOX2*, *CEBPA*, and *ZNF71* [25] (Figure 5B). This TF regulon participates in the *Wnt*-mediated tumorigenesis, and multiple intracellular intrinsic and innate immune systems [26,27]. Also, the TIMER2.0 program analysis results showed that increased *SOX4* expression was positively associated with the increased expression of multiple oncogenes, including *EGFR*, *FN1*, *KRAS*, and *mTOR* (Figure 5C). *CTNNB1*/*SOX4* expression in CRC tumors was significantly associated with tumor-infiltrating CAFs (Figure 5D).

### 2.6. In Silico Molecular Docking Analysis of Sitagliptin against CD24 with Sitagliptin

After establishing the *DDP4*′s connection with a *CD24*/*SOX4*-centered signaling hub, we decided to explore whether the *DPP4* inhibitor, sitagliptin, could target colon CSCs. We applied molecular docking simulations using the Autodock vina tool, which showed the high binding abilities of sitagliptin when docked with *DPP4* and *CD24*. The Gibbs free energy obtained from the sitagliptin–*DPP4*/*CD24* (PDB:2ONC) complex exhibited a binding affinity of (Δ = −7.4 and Δ = −9.6 kcal/mol), respectively (Figure 6A,B). Our careful analysis of the interactions between sitagliptin and *DPP4* and *CD24* revealed a higher number of van der Waals forces, salt bridge, pi-pi stacked, pi-pi shaped, pi-alkyl, and most importantly, two conventional hydrogen bonds and their minimal distance constraints, as shown in the accompanying table Interestingly, sitagliptin is a *DPP4*-specific inhibitor, and our docking analysis predicted that sitagliptin was bound to *CD24* with an even higher affinity than *DPP4*.

### 2.7. Molecular Targeting of β-Catenin and SOX4 by Sitagliptin

Next, we performed a molecular docking analysis for key molecules in the *CD24*/*SOX4*-centered signaling hub, *SOX4*, and *β-catenin* by association. We demonstrated that sitagliptin formed stable interactions with *β-catenin* (or *CTNNB1*) (PDB: IJPW) and *SOX4* (PDB:3U2B). Specifically, *β-catenin* and *SOX4* formed a stable complex with sitagliptin exhibiting Gibbs free energies of Δ = −7.4 and Δ = −9.1 kcal/mol, respectively (Figure 7A,B). We used the discovery studio (BIOVIA) to analyze our results further. We identified interactions between sitagliptin, which displayed van der Waals interactions, halogens, pi-cation, alkyl, three conventional hydrogen bonds, and the minimum binding distances, as shown in the accompanying table below.

### 2.8. Sitagliptin Decreases the Viability of CRC Cells

The efficacy of sitagliptin on the viability of CRC cells was evaluated to confirm the predicted outcomes. The results indicated that sitagliptin reduced the viability of DLD-1 and HCT116 cells with IC50 values of 140 µM and 270 µM, respectively (Figure 8A). Additionally, sitagliptin was found to have inhibitory effects on migration, colony and sphere formation of DLD-1 and HCT116 cells (Figure 8B–E).

### 2.9. Sitagliptin Synergistically Enhanced the Anti-Tumorigenic Activities of 5FU in CRC Cells

The synergetic combination index (CI index) in the range of 0.06–6.29 and 0.65–6.75 was obtained for DLD-1 and HCT116 (Figure 9A,B), thus indicating that sitagliptin and 5FU act synergistically to suppress the viability of CRC. Furthermore, colony formation assay demonstrated that, the combination of sitagliptin and 5FU significantly decreased the colony formation of DLD-1, as compared with 5FU single treatment (Figure 9C). Moreover, the combination treatment also suppressed the sphere forming ability of DLD-1 and HCT116, as compared with single treatment (Figure 9D). Accordingly, western blot analysis following combination of sitagliptin and 5FU, exhibited more suppressive effects on *DPP4*, *CD24*, *β-catenin*, and *SOX4* centered signaling hub, and GAPDH was used as internal control, as compared with a single treatment (Figure 9E).

## 3. Discussion

Despite the significant advances in treatment modalities, colorectal cancer (CRC) remains one of the leading malignancies, with high numbers of reported cases and causes of death globally [28]. The major causes of CRC-related deaths are associated with cancer recurrence and metastasis. Most studies have shown that approximately 60% of CRC patients are expected to develop metastasis [29,30]. Although the current approaches for the treatment of CRC have improved significantly over the last decade, the overall survival rate for most patients is still under 5 years. Several studies have identified subpopulations of CRC that are more resistant to therapeutics, such as chemotherapy and radiotherapy [5,6,8]. Chemoresistance remains an obstacle in CRC patients and is a major factor in Fluorouracil (5FU) therapeutic resistance in CRC patients [31]. 5FU-based therapy comes with severe adverse effects, and many patients eventually develop resistance [32,33]. Thus, there is an urgent need to identify novel and reliable biomarkers associated with cancer stemness and drug resistance for drug development in CRC. Dipeptidyl peptidase 4 (*DPP4*) oncogene is frequently expressed in cancer tissue and has been reported to promote cancer development, progression, and metastasis in CRC, making it a potential and novel therapeutic target [16,17,18]. Sitagliptin, an anti-diabetic drug with cytotoxic activities, was approved by the FDA in 2006 as an inhibitor of *DPP4* [15,34,35]. Recently, studies have shown that sitagliptin inhibits mCRC in vivo [9,14]. In the present study, we evaluated the anticancer activities of sitagliptin as a target for *CD24*/*CTNNB1*/*SOX4*, which were previously demonstrated to promote CRC progression, stemness, and metastasis [36,37,38]. Studies have shown that *DPP4* modulates the T-cell mechanism and promotes tumor initiation [39,40,41]. Moreover, others have demonstrated the significant role of *DPP4* in CRC development, progression, and association with therapeutic resistance [15,42,43,44].

*DPP4* has also been demonstrated to interact with the tumor microenvironment [13,14,15], suggesting its essential role in CRC. Herein, we explored computational simulations to identify potential target genes upregulated in CRC. We performed transcriptomics data analysis from GEO, which revealed 59 upregulated oncogenes in CRC tissues, compared with normal tissue; interestingly, among the top genes expressed were *CD24*/*CTNNB1*/*SOX4*. To validate the expression of this oncogenic signature in CRC, we utilized the GEPIA2 bioinformatics tool, which confirmed the overexpression of these oncogenes as expected. To validate further, we explored the human protein atlas to identify the expression of these oncogenes at a tissue level and found high expression levels of these genes in CRC tissues as compared with tissues with medium-staining intensities displayed on each tissue. To further analyze, we found that high expressions of *CD24*, *CTNNB1*, and *SOX4* oncogenic signatures were associated with a shorter overall survival rate in CRC, as shown on the Kaplan–Meier plots (Figure 3D–F).

To analyze the possible protein–protein interactions (PPI) among *DPP4*, *CTNNB1*, *CD24*, and *SOX4* oncogenic signatures, we applied the STRING analytic tool to try to connect *DPP4* with *CD24*/*CTNNB1*/*SOX4* signaling. We identified high-confidence interaction between *DPP4* and *CD24*/*CTNNB1*/*SOX4*; these interactions were associated with gene neighborhood, gene fusion, gene co-occurrence, and co-expression among the genes, as well as enriched gene ontology involving biological processes (BP), and KEGG pathways, which, interestingly, were associated with immune response, hence suggesting the potential inhibitory effects of the *DPP4* inhibitor on *CD24*/*CTNNB1*/*SOX4* oncogenic signaling.

These findings prompt us to predict the protein–ligand interaction further, using molecular docking to evaluate the potential binding affinities of sitagliptin with *DPP4*/*CTNNB1*/*CD24*/*SOX4* oncogenic signatures. The results from the in silico molecular docking analysis revealed unique binding energy between sitagliptin with *DPP4*, *CTNNB1*, *CD24*, and *SOX4* with Gibbs free energy values of (−7.4 kcal/mol, −7.4 kcal/mol, −9.6 kcal/mol, and −9.1 kcal/mol); these observations strongly suggested that sitagliptin could form a stable complex with *DPP4*/*CTNNB1*/*CD24*/*SOX4* oncogenic signature. We further investigated sitagliptin’s therapeutic potential using in vitro models and found that sitagliptin and 5FU acted synergistically to suppress the viability of CRC cells and decreased migration ability, colony, and tumorsphere formation compared with 5FU single treatment. Collectively, these results suggest that a combination of sitagliptin and 5FU could serve as a novel strategy to improve the efficacy of 5FU in colorectal cancer.

## 4. Materials and Methods

### 4.1. Microarray Data Acquisition and Identification of Differentially Expressed Genes in CRC

The transcriptomic data were downloaded from the Gene Expression Omnibus (GEO) database (https://www.ncbi.nlm.nih.gov/geo/ accessed on 4 April 2023) [45], accession numbers GSE21510 [46], GSE32323 [47], GSE110223 [48], and GSE110224 [48]; microarray data processing were processed and analyzed using GEO2R, an R-based web application that identifies differentially expressed genes (DEGs) [45]. The determination of the correct fold change cut-off value in the transcriptomic data analysis remains a great challenge [49]; herein, we considered the Benjamin–Hochberg accuracy methods for *p*-value adjustment of the false discovery rate (FDR), as previously reported [50]. We considered *p* value less than 0.05 and fold change greater than 1.5 as the criteria for DEG selection to the upregulated genes in tumor samples compared with normal adjacent samples. Bioinformatics and evolutionary genomic diagram was used for constructing Venn diagrams to visualize overlapping (http://bioinformatics.psb.ugent.be/webtools/Venn/ accessed on 4 April 2023) [51].

### 4.2. Identification of Differentially Expressed Genes (DEGs) in CRC

The messenger-RNA (mRNA) levels of differentially expressed genes (DEGs) of in tumorous versus normal tissues from various cancers from The Cancer Genome Atlas (TCGA) database were analyzed using GEPIA2 (http://gepia2.cancer-pku.cn/ accessed on 4 April 2023) bioinformatics software with default settings. To validate the expression levels of identified DEGs in CRC [52], to validate these results, we explored the Human Protein Atlas (HPA) database for immunohistochemistry (IHC) (https://www.proteinatlas.org/ accessed on 4 April 2023) to compare expression levels between tumor samples and normal samples.

### 4.3. Correlation Analysis of Immune Cell Infiltration and CD24/CTNNB1/SOX4 Expressions

Correlations between *CD24*/*CTNNB1*/*SOX4* expressions and tumor infiltration levels were analyzed with the Tumor Immune Estimation Resource (TIMER 2.0, http://timer.cistrome.org/ accessed on 5 April 2023) [43]. We mainly analyzed correlations of *CD24*/*CTNNB1*/*SOX4* and infiltration of cancer-associated fibroblasts (CAFs) in LUAD. In addition, we also analyzed mutations of *CD24*/*CTNNB1*/*SOX4* in CAFs using the mutation module from TCGA clinical outcomes in the TIMER 2.0 algorithm. For further analysis, we determined distributions of BIRC5/VEGFA/HIF1A expressions in LUAD across different molecular subtypes including wound healing, interferon (IFN)-gamma dominant, inflammation, lymphocyte, and transforming growth factor (TGF)-β subtypes using TISIDB (http://cis.hku.hk/TISIDB/ accessed on 5 April 2023) [44].

### 4.4. Correlation Analysis between CD24/CTNNB1/SOX4 Oncogenic Signatures

In a further analysis, we applied STRING (https://string-db.org/ accessed on 26 May 2023), a network functional analysis tool, used to predict a protein–protein interaction (PPI) between genes, and gene ontology (GO) enrichment analysis. [53,54]. The results acquired from the analysis displayed the following network statistics; number of nodes (4), number of edges (5), average node degree (2.5), average local clustering coefficient (0.8) and PPI enrichment *p*-value (8.97 × 10^−5^). For further analysis, we utilized the DAVID, a functional annotation clustering tool (https://david.ncifcrf.gov/ accessed on 26 May 2023), to measure gene–gene similarity [55], then applied the funrich: *functional enrichment analysis tool* (http://www.funrich.org/ accessed on 26 May 2023), to analyze the gene ontology (GO), including biological functions and KEGG: Kyoto encyclopedia of genes and genome pathways [56].

### 4.5. Interpretation of Gene Co-Expression in CD24/CTNNB1/SOX4 Genes Network

Interpretation of gene expression network was analyzed using the Network Analyst 3.0 tool (https://www.networkanalyst.ca/ accessed on 28 May 2023), a comprehensive analytical visual platform which integrates PPI networks and gene co-occurrence networks and interprets gene expression networks. Herein, we used Enrichment Map, a sub-tool of network analyst, with a Bipartite view, to determine the enrichment of co-expressed genes in KEGG pathways [57,58,59].

### 4.6. Distribution and Expression of CD24/CTNNB1/SOX4 in Colon Tissue

A clear understanding of single-cell RNA expression in a specific cell type in colon tissue is crucial for the investigation of specific and potential cancer biomarkers that may promote tumorigenesis [60]. We used the HPA database (https://www.proteinatlas.org/ accessed on 28 May 2023) to retrieve the cell-type atlas showing the single-cell RNA sequencing (scRNA-seq) data of *DPP4*, *CTNNB1*, *CD24*, and *SOX4* expression in colon tissue [61]. The expression of each gene in the specified cell types was explored using the interactive UMAP plots and bar chart [62].

### 4.7. In Silico Analysis of Molecular Docking of Receptors and Ligands

To predict the potential inhibitory activities of sitagliptin and sulfasalazine, we applied computational simulation using the silico molecular docking of the two compounds in complex with *DPP4*, *CTNNB1*, *CD24*, and *SOX4* oncogenes, and further compared the binding affinities of both compounds when bound to the abovementioned signatures. The three-dimensional (3D) structures of sitagliptin (CID:4369359), molecular weight (MW: 407.31 g/mol), and sulfasalazine (CID:5339, MW: 398.45 g/mol) were retrieved from PubChem database as SDF files, which were later converted to PDB file format using Pymol visualization software (https://pymol.org/2/ accessed on 2 July 2023), and subsequently, converted to PDBQT format using autodock software (version 1.5.6). (https://autodock.scripps.edu/ accessed on 2 July 2023). The crystal structures of *DPP4* (PDB:2ONC), *CTNNB1*(PDB:1JDH), *CD24* (PDB:3SGD), and *SOX4* (PDB:3U2B), were downloaded from the Protein Data Bank (PDB) (https://www.rcsb.org/ accessed on 2 July 2023). Molecular docking analysis was performed using autodock software (version 1.5.6), and further 3D visualization and interpretation were analyzed using PYMOL and discovery studio software (version 2.5.7).

### 4.8. Cell Culture and Reagents

The DLD-1 and HCT116 human colon cancer cell lines were acquired from the American Type Culture Collection (ATCC) based in Manassas, VA, United States of America. In brief, each cell line was cultured and subsequently passaged at 90% cell confluence using Dulbecco’s Modified Eagle Medium (DMEM) sourced from Invitrogen, Life Technologies in Carlsbad, CA, USA. The cells were then stored under standard incubator conditions, maintaining a temperature of 37 °C with 5% humidified CO_2_.

### 4.9. Cell Viability Assay

We assessed cell viability using sulforhodamine B (SRB reagent from Sigma-Aldrich, Taipei, Taiwan, following established protocols) [63]. HCT116 and DLD-1 cells at confluency of more 90% were plated at 7000 cells/well in 96-well plates for 24 h. After a 48 h treatment with sitagliptin, we added 10% TCA, refrigerated at 4 °C for an hour, washed twice with distilled water, and stained with 0.4% SRB for 30 min at room temperature. After two washes with 1% acetic acid, plates air-dried overnight. The protein-bound stain was dissolved in a 20 mM Tris-buffer solution on an orbital shaker for 15 min, and absorbance was measured at 560 nm wavelength using a microplate reader (Molecular Devices, Sunnyvale, CA, USA).

### 4.10. Cell Migration Assay

CRC cells (HCT116 and DLD-1) were seeded at a density of 10^5^ cells in 100 μL of media per well in two-well plates with a silicon insert. After 24 h of incubation, the media were removed, and the insert was gently taken out. Fresh media (1.5 mL) containing sitagliptin (11.25 μM) were added. Wound images were captured immediately (0 h) and after 24 h using an Olympus CKX53 Cell Culture Microscope. Wound closure was quantified with NIH ImageJ software (version 1.8.0) (https://imagej.nih.gov/ij/ accessed on 2 July 2023).

### 4.11. Colony-Formation Assays

To evaluate how sitagliptin treatment influences colony formation in colon cancer cell lines, we conducted a colony formation assay based on Franken et al.’s protocol [64]. Specifically, 500 cells were seeded in Corning^®^ 6-well plates (Sigma-Aldrich) and treated with sitagliptin at concentrations equivalent to the 40% inhibitory concentration (IC40) values for HCT116 and DLD-1. The cells were allowed to grow for a minimum of one week. Colony quantification was performed using a Cell3iMager neo-scanner, and inhibitory effects on colonies were expressed as percentages relative to control colonies.

### 4.12. Immunoblot Analysis

The DLD-1 and HCT116 cell line groups, both treated with sitagliptin along with the control group, underwent trypsinization for harvesting. Total protein lysates from treated and untreated cells were collected using RIPA buffer. Subsequently, 20 µg of total lysates were separated via SDS-PAGE using the Mini-Protean III system (Bio-Rad, New Taipei City, Taiwan) and transferred onto polyvinylidene difluoride membranes with the Trans-Blot Turbo Transfer System (Bio-Rad) [65]. Membranes were then incubated overnight in the refrigerator at −4 °C with primary antibodies, followed by a 1 h incubation with secondary antibodies the next day, list of antibodies are shown on Table 1. Enhanced chemiluminescence (ECL) detection kits (ECL kits; Amersham Life Science, California, CA, USA) were used to detect proteins of interest, and BioSpectrum^®^ Imaging System (Upland, CA, USA) was employed for image capture and analysis.

### 4.13. Tumor-Sphere Formation Ability

HCT116 and DLD-1 tumor spheres were formed in serum-deprived conditions following Johnson et al., 2013 [66]. Specifically, 2000 colon cancer cells were seeded per well in six-well ultra-low-attachment plates (Corning, Corning, NY, USA) using serum-free media. After 7 days, tumor spheres, characterized as dense, non-adherent spheroid-like masses with a diameter over 50 µm, were counted using an inverted phase-contrast microscope.

### 4.14. Data Analysis

Pearson’s correlations were used to assess correlations between *DPP4*/*CTNNB1*/*CD24*/*SOX4* expressions in multiple cancer types. The statistical significance of DEGs was evaluated with * *p* < 0.05 was accepted as being statistically significant. The Kaplan–Meier curve was employed to present patient survival in different cancer cohorts; Western blot quantification was evaluated using Image J (Version 1.53t).

## 5. Conclusions

In summary, we identified overexpression of *CD24*, *CTNNB1*, and *SOX4* signatures in CRC, revealing their association in promoting tumorigenic properties, cancer stemness, and resistance to therapy. In addition, the in silico molecular docking analysis revealed high binding energies between sitagliptin *DPP4*, *CTNNB1*, *CD24*, and *SOX4*, predicting its potential inhibitory functions against these targets. Subsequently, we validated the above findings in the CRC cell line model, where the combination of sitagliptin and 5FU suppressed the migration, colony, and tumorsphere formation of DLD-1 and HCT116 cells, accompanied by the reduced *DPP4*/*CTNNB1*/*CD24*/*SOX4* expression.

## Figures and Tables

**Figure 1 ijms-25-00609-f001:**
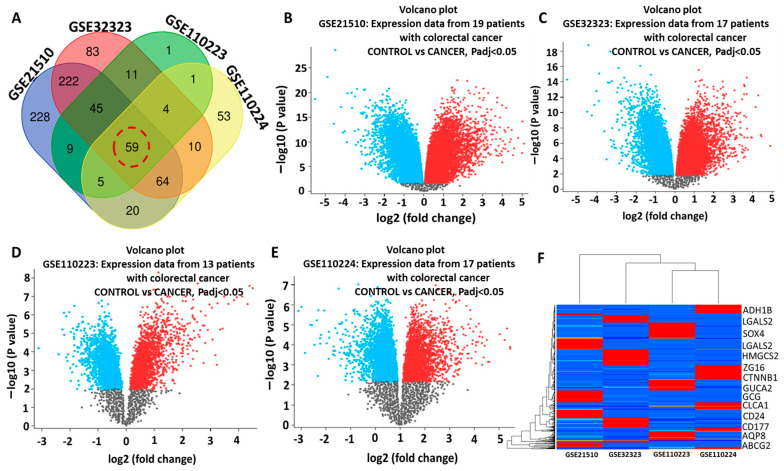
Identification of potential differentially expressed genes (DEGs) related to colorectal cancer (CRC). (**A**–**D**) Volcano plots showing differentially expressed genes (DEGs) in the array datasets between CRC patients and the control groups. (**E**) Venn diagram showing 59 overlapping upregulated DEGs in each dataset from all CRC datasets. (**F**) Heat maps of the overlapping upregulated DEGs obtained from the four datasets. Among the 59 upregulated DEGs were (ADH1B, LGALS2, *SOX4*, *LGALS2*, *HMGCS2*, *ZG16*, *CTNNB1*, *GUCA2*, *GCG*, *CLCA1*, *CD24*, *CD177*, *AQP8*, and *ABCG2*).

**Figure 2 ijms-25-00609-f002:**
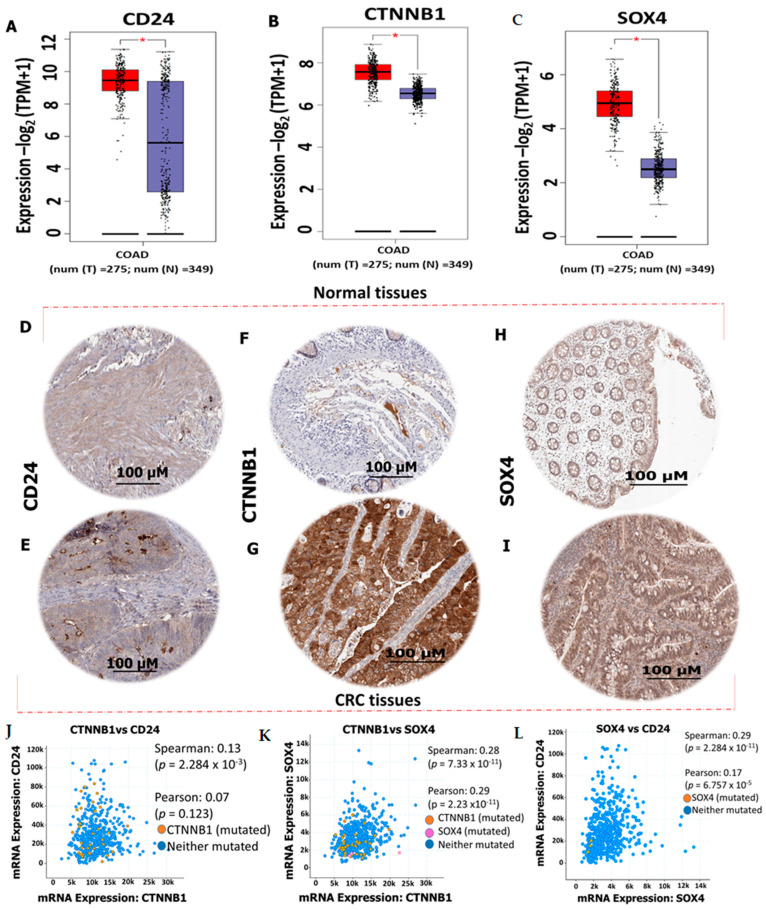
*CD24*/*CTNNB1*/*SOX4* oncogenes are overexpressed in CRC tissues as compared with normal tissues. (**A**–**C**), high expression levels of *CTNNB1*, *CD24*, and *SOX4* in CRC. HPA staining intensity was analyzed based on medium staining (**D**–**I**) intensities of *CD24*, *CTNNB1*, and *SOX4* in normal colon tissues compared with CRC tissues**.** Overexpression of *CD24*/*CTNNB1*/*SOX4* is associated with poor prognosis in CRC. Co-expression of (**J**) *CTNNB1* with *CD24* (**K**) *CTNNB1* with *SOX4*, (**L**) *SOX4* with *CD24* in CRC. The staining quality was <75%, and * *p* < 0.05 was considered statistically significant.

**Figure 3 ijms-25-00609-f003:**
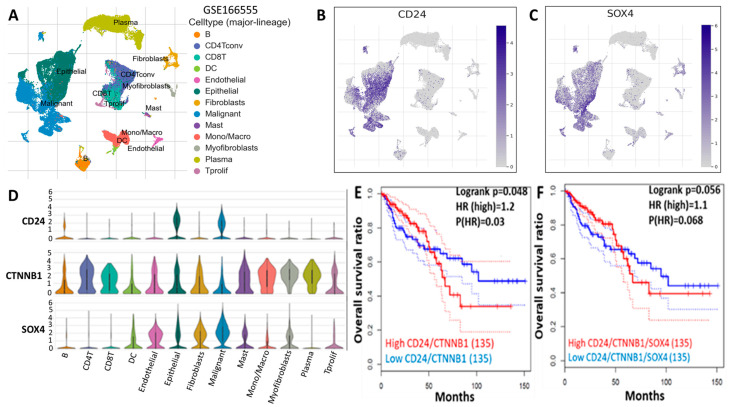
*CD24*/*SOX4* oncogenes are overexpressed in CRC and correlated with poor prognosis (**A**–**F**).

**Figure 4 ijms-25-00609-f004:**
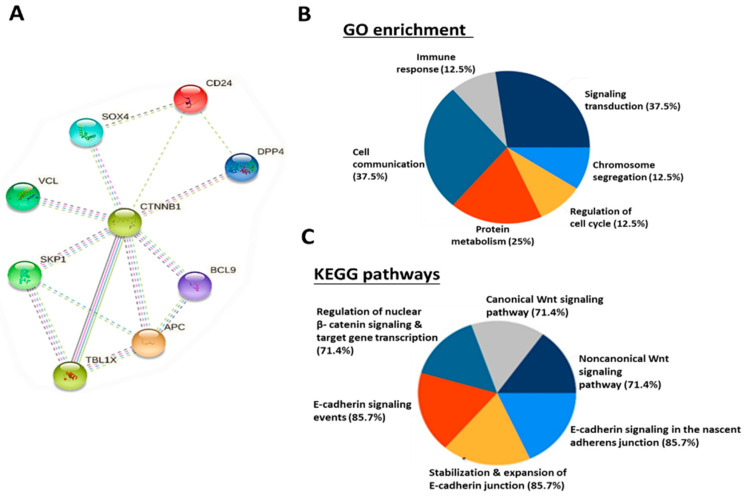
PPI networks centered around *CD24*/*CTNNB1*/*SOX4*. Interactions are shown after considering the gene neighborhood, gene fusion, gene co-occurrence, and co-expression of (**A**) *DPP4* with *CTNNB1*, *CD24* with *CTNNB1*, *SOX4* with *CTNNB1*, *DPP4* with *CD24*, and *SOX4* with *CD24*. (**B**) Six biological processes associated with *CD24*/*SOX4* oncogenic signature-correlated gene clusters. (**C**) KEGG pathway showing six affected pathways by *CD24*/*SOX4*/*DPP4*/*CTNNB1* oncogenic signatures, which were significantly associated with several functions, represented in percentages (%).

**Figure 5 ijms-25-00609-f005:**
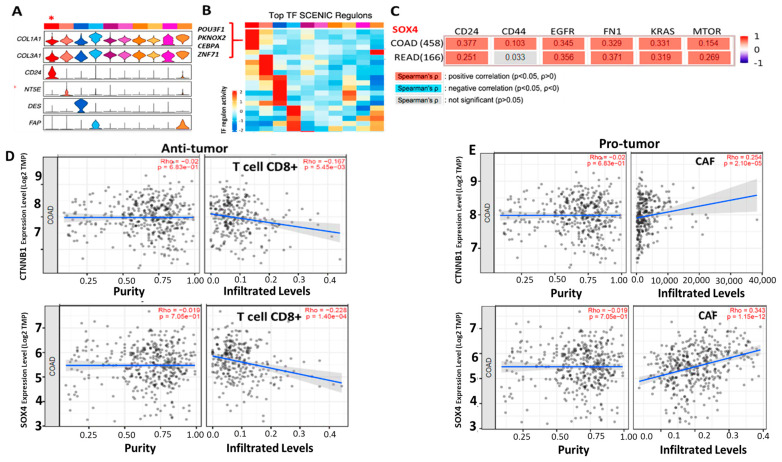
*CD24*/*CTNNB1*/*SOX4*-centered signaling in CRC tumor cells and cancer-associated fibroblasts predicts poor survival in CRC patients. (**A**) Violin plots demonstrate the heterogeneity of CAFs. A red asterisk marks the *CD24*+ CAFs in the first column. (**B**) A heatmap indicates the top transcriptional factor (TF) SCENIC regulons. The top-ranking TFs associated with *CD24*+ CAF are listed in red. Data in (**A**,**B**) were adapted from Qi et al. [25]. (**C**) A table demonstrates *SOX4*-co-expressing oncogenes. The number in each box indicates Spearman’s correlation coefficient. (**D**,**E**) Expression level versus CAF-infiltration level plots for *CTNNB1*/*SOX4* in TCGA CRC databases. The results were generated using TIMER2.0 software.

**Figure 6 ijms-25-00609-f006:**
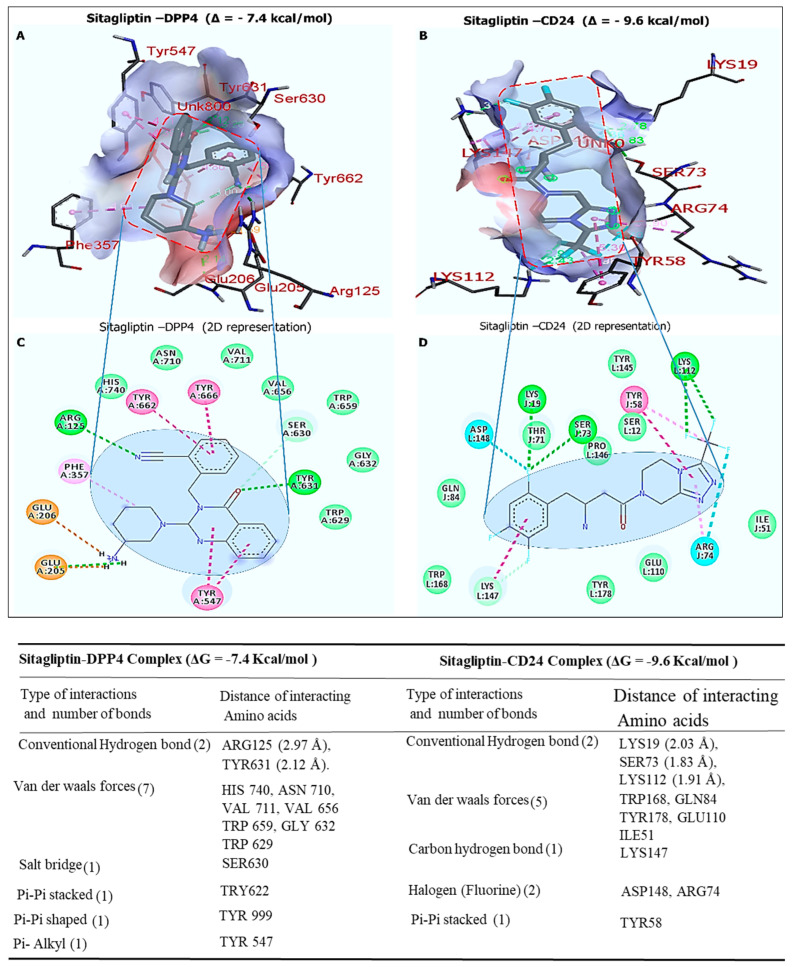
Molecular docking profiles of sitagliptin with the *DPP4*/*CD24* oncogenes. (**A**,**B**) Three-dimensional representation showing strong binding energies of sitagliptin with *DPP4* at Δ = −7.4 kcal/mol and of *CD24* at Δ = −9.6 kcal/mol, respectively. (**C**,**D**) Two-dimensional visualization of docking results of interactions between sitagliptin and *DPP4* revealed two conventional hydrogen bonds and their minimal distance constraints, including LYS19 (2.03 Å), SER73 (1.83 Å), and LYS112 (1.91 Å) for *DPP4* and ARG125 (2.97 Å) and TYR631 (2.12 Å) for *CD24*. The accompanying table shows several interactions, including amino acids, van der Waals forces, carbon–hydrogen bonds, halogens, and pi-pi stacked.

**Figure 7 ijms-25-00609-f007:**
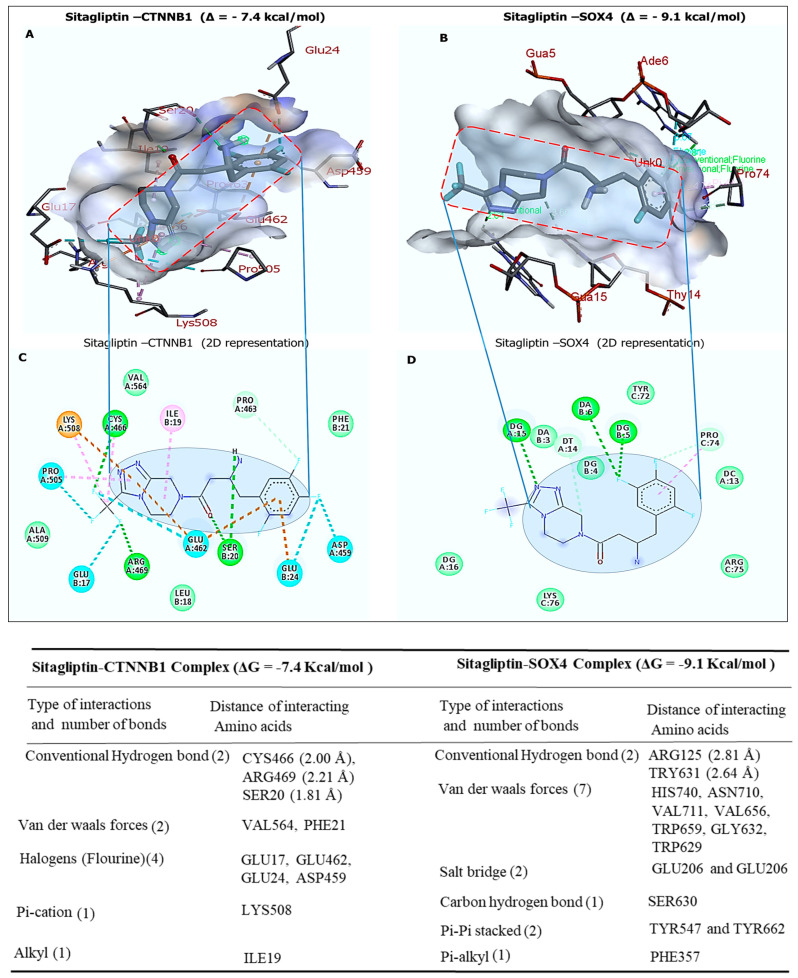
Molecular docking profiles of sitagliptin with *SOX4* and *CTNNB1*(*β-catenin*). (**A**,**B**) Three-dimensional representation showing strong interactions between sitagliptin with *CTNNB1* at Δ = −7.4 kcal/mol and of *SOX4* Δ = −9.1 kcal/mol, respectively (**C**,**D**) Two-dimensional visualization of docking results of conventional hydrogen bonds with their short binding distances of CYS466 (2.00 Å), ARG469 (2.21 Å), and SER20 (1.81 Å), when *CTNNB1* bond with sitagliptin and ARG125 (2.81 Å) and TRY631 (2.64 Å), for *SOX4* in complex with sitagliptin. The accompanying table shows several interactions, including amino acids, van der Waals forces, carbon–hydrogen bonds, halogens, and pi-pi stacked.

**Figure 8 ijms-25-00609-f008:**
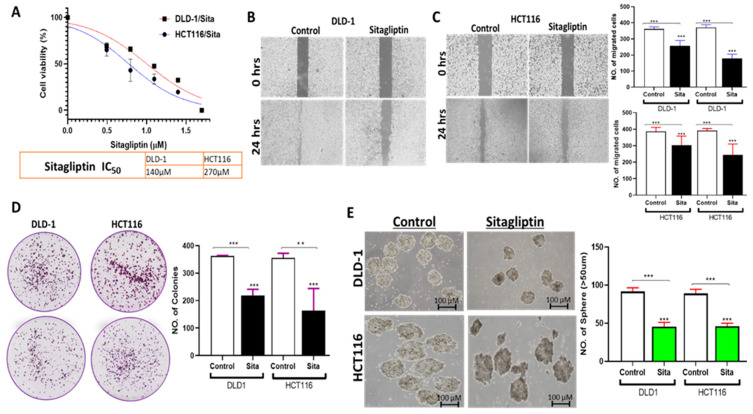
Sitagliptin displayed inhibitory effects on the viability of CRC Cells (**A**). The 50% inhibitory concentration (IC50) values for HCT116 and DLD-1 cell lines. (**B**,**C**) Sitagliptin suppressed the migration ability (**D**) colonies and (**E**) sphere formation of DLD-1 and HCT116 cells. Images of the colonies and spheroids are shown in the left panel, with quantification of the results presented in the right panel. Statistical significance is denoted by ** (*p* < 0.05) and *** (*p* < 0.01).

**Figure 9 ijms-25-00609-f009:**
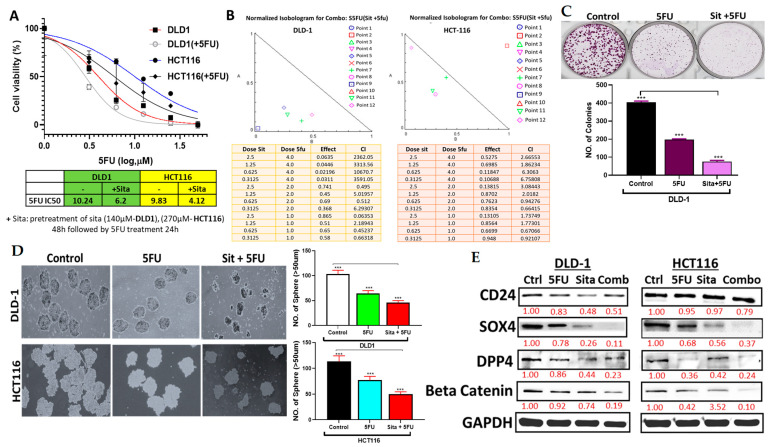
Sitagliptin treatment reduced tumorigenic properties of CRC cells and enhanced 5FU efficacy. (**A**) Sita enhanced 5FU efficacy in both DLD-1 and HCT116 cell lines. IC50 values are shown. (**B**) Combination of sitagliptin and 5FU, improved efficacy of 5FU in CRC CI index of sitagliptin and 5FU combination treatment in DLD-1 and HCT116 cells. The combination index (CI) <1 defines synergism; CI of 1 indicates additive effect while >1 indicates antagonism. The DLD-1 and HCT116 cell lines were exposed to a combination treatment of 5FU and sitagliptin for 7 days. (**C**) Images of colonies and (**D**) spheroids were taken, followed by counting. Western blotting analysis showed significant suppressive effects of 5FU with sitagliptin on *DPP4*, *CD24*, *β-catenin*, and *SOX4* expression compared with 5FU alone. GAPDH was used as an internal control. (**E**) Statistical significance is denoted by *** (*p* < 0.01).

**Table 1 ijms-25-00609-t001:** List of antibodies.

Target	Dilution	Company and Catalog No.	Predicted MW (kDa)
GAPDH	1:5000	Proteintech (Rosemont, IL, USA), GAPDH, Rabbit mAb, 10494-1-AP	36
β-catenin	1:1000	Cell Signaling (Danvers, MA, USA), β-catenin (63) Rabbit mAb, #9582	92–100
CD24	1:1000	Proteintech, CD24, Rabbit mAb, 10600-1-AP	30–70
DPP4	1:1000	Proteintech, DPP4, Rabbit mAb, 10940-1-AP	55–60
SOX4	1:500	Proteintech, SOX4, Rabbit mAb, 27414-1-AP	47
2nd Antibodies	1:5000	Proteintech, Biotin-conjugated Affinipure Goat Anti-Rabbit IgG(H+L), SA00004-2	

## Data Availability

The datasets generated and/or analyzed in this study are available on reasonable request.

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
