# Peer review of "Preclinical Repurposing of Sitagliptin as a Drug Candidate for Colorectal Cancer by Targeting CD24/CTNNB1/SOX4-Centered Signaling Hub"

_ijms, 2024, doi:10.3390/ijms25010609_

Round 1

Reviewer 1 Report

Comments and Suggestions for Authors

Although the results presented by Shih et. Al are interesting I cannot accept the manuscript in the present form. The main problem is that authors did not describe the in vitro experiments at all. There is no information from where the cell lines were purchased and how were cultured. What was the concentration and the time of treatment with sitagliptin? What was the final concentration of seeded cells? In case of the western blot  analysis there is no information regarding the concentration of antibody, cat number etc. The figure 9 E is not properly labeled, it is not clear which images relate to DLD1 and which to HCT116. Also, in the raw data Authors included cropped images, please attache all blot images. In case of SOX4 I am quite suspicious weather left part of the image correspond to left part of 9E image. There is no similar pattern. Which program was used to evaluate densitometry?

In case of data analysis there is no information regarding the statistic test that was used to evaluate analysis.

As an additional test to proof that combination of sitagliptin and 5FU have a better effect on colorectal cancer elimination I would suggest to perform  simple test - Annexin V test on single drugs and combination of the drugs.  

Minor issues:

Line 84: double space bar after CSC

Line 143: space bar in front of comma

Reviewer 2 Report

Comments and Suggestions for Authors

The manuscript by Shih and co-workers identifies a signaling hub centered on CD24/CTNNB1/SOX4 in colorectal (CRC) cancer and explores in vitro the possibility that a marketed drug for type 2 diabetes could be repurposed to interfere with such signaling hub

The manuscript is interesting for the oncology community, many experiments have been conducted generating data and a hypothesis, however

Major comments

While the findings are interesting, the possibility of translating them to the clinics is not adequately discussed. Authorized posology for sitagliptin in type 2 diabetes is 100 mg per day by mouth. Is the drug exposure obtained with this posology close to IC50s of 14 to 17 micromolar reported in the manuscript?

Is there reason to think that sitagliptin may synergize with other chemotherapeutic agents commonly used in CRC (e.g., oxaliplatin)?

Minor comments

Introduction

Is there a real need to cite both refs 2 and 3?

Line 50: please remove “and”

Lines 54-55: is the expression of the stemness markers referred to CRC or cancer more in general? Is there a marker profile of CSC in CRC?

Line 69: please insert a space between CRC and [16-18]

Line 82:  where is the preliminary data section?

Results

Fig 1 legend: there are 2 full stops at the end of the legend on line 102

Lines 100-102: why are the listed genes in brackets?

Line 105: use “;” instead of “,”?

Line 106: use “:” instead of “,”?

Fig. 2 legend: please double check the whole legend, in particular, there is no reference to panels J, K and L

Line 134: please add “survival” after overall

Fig. 3 legend: there is no description of the panels A to F

Line 144: fig 2A is instead fig 4A?

Line 148: fig 2B is instead fig 4B?

line 152: fig 2C is instead fig 4C?

 Fig 5 legend: there is no description of panel E

Figure 6 legend: there are 2 full stops at the end of the legend on line 207

Fig 9 legend please double check the whole legend as description of results does not correspond to indicated panels

Discussion

Line 272: is it “expressed cancer tissue” or “expressed in cancer tissue”?

Line 286: is it “among the top genes expressed” or “among the top gene expressed were”

Line 296: “We” should be “we”

Materials and Methods

There is no description of the cell lines used and of experiments such as colony formation, sphere formation

Reviewer 3 Report

Comments and Suggestions for Authors

The presented study is globally well described and the manuscript is reasonably well written.

Most of the study is based on in silico analysis and interaction / structural compatibilities.

One part is missing, related to the methods for in vitro experiments (figures 8 and 9), which is mandatory for results evaluation.

Round 2

Reviewer 1 Report

Comments and Suggestions for Authors

I am very sorry, however, I can not accept the response regarding the western blot image for SOX4 for the HCT116 cell line since it is a different pattern that the Authors show in the manuscript. I can reccommend the manuscript for publication only if the Authors present original image ( which has the same pattern). Here even after light merging, as the Authors explain, it should be that the first line is the strongest one (as in the manuscript) and looking at the raw data, the first line is not visible. Additionally, the raw data images should be properly labeled. The Authors should have replicates for western blot experiment - then it should not be a problem to present different images compatible with raw data.

Additional comments:

- Line 442: CO2 instead CO2

- Line 478: Imminoblot instead immunoblo

- Line 484:  "Membranes were then incubated
overnight at -4°C with primary antibodies, followed by a 1-hour incubation with secondary antibodies the next day."
Was the incubation performed at -4°C?
Also, the concentration and catalog number for primary and secondary antibodies are not provided.

Reviewer 2 Report

Comments and Suggestions for Authors

The authors have improved the quality of presentation, however one major concern is that the results cannot be translated to the clinics. In the original version, IC50 for sitagliptin were 16.95 and 14.04 micromolar on DLD-1 and HCT116 cell lines, respectively. In the revised version, the IC50 are 140 and 270 micromolar, an exposure that is highly unlikely to be achieved in humans following assumption of sitagliptin at authorized dose (100 mg/day)

Please check if there is a typo and the IC50 are actually lower than reported

the above are IC50 of direct citotoxicity of sitagliptin on cancer cells (both non-CSC, the vast majority, and CSC, very few if any), it is probably more important to address the exposure to sitagliptin required to inhibit CSC only, or the minimum exposure that affords synergism with 5-FU against cancer cells, please provide such evaluation

minor comments:

please move figure 8 legend that seems to be misplaced

the revision of figure 9 legend is not satisfactory: panel A and B are likely to be presented together, C-D should be on line 294 of the legend

Reviewer 3 Report

Comments and Suggestions for Authors

manuscript is now fine

Round 3

Reviewer 1 Report

Comments and Suggestions for Authors

I did not notice the information redarging the concentration and catalog number for primary and secondary antibodies in the revised manuscript.

Please add it.

Afterwards, I suggest the manuscript for publication.
